# Auxin Metabolome Profiling in the Arabidopsis Endoplasmic Reticulum Using an Optimised Organelle Isolation Protocol

**DOI:** 10.3390/ijms22179370

**Published:** 2021-08-29

**Authors:** Ludmila Včelařová, Vladimír Skalický, Ivo Chamrád, René Lenobel, Martin F. Kubeš, Aleš Pěnčík, Ondřej Novák

**Affiliations:** Laboratory of Growth Regulators, Institute of Experimental Botany of the Czech Academy of Sciences, and Faculty of Science, Palacký University, Šlechtitelů 27, CZ-78371 Olomouc, Czech Republic; ludmila.vcelarova@upol.cz (L.V.); vladimir.skalicky@upol.cz (V.S.); ivo.chamrad@upol.cz (I.C.); rene.lenobel@upol.cz (R.L.); martin.kubes@uhk.cz (M.F.K.); ondrej.novak@upol.cz (O.N.)

**Keywords:** endoplasmic reticulum, auxin, subcellular fractionation, density gradient centrifugation, mass spectrometry

## Abstract

The endoplasmic reticulum (ER) is an extensive network of intracellular membranes. Its major functions include proteosynthesis, protein folding, post-transcriptional modification and sorting of proteins within the cell, and lipid anabolism. Moreover, several studies have suggested that it may be involved in regulating intracellular auxin homeostasis in plants by modulating its metabolism. Therefore, to study auxin metabolome in the ER, it is necessary to obtain a highly enriched (ideally, pure) ER fraction. Isolation of the ER is challenging because its biochemical properties are very similar to those of other cellular endomembranes. Most published protocols for ER isolation use density gradient ultracentrifugation, despite its suboptimal resolving power. Here we present an optimised protocol for ER isolation from *Arabidopsis thaliana* seedlings for the subsequent mass spectrometric determination of ER-specific auxin metabolite profiles. Auxin metabolite analysis revealed highly elevated levels of active auxin form (IAA) within the ER compared to whole plants. Moreover, samples prepared using our optimised isolation ER protocol are amenable to analysis using various “omics” technologies including analyses of both macromolecular and low molecular weight compounds from the same sample.

## 1. Introduction

The endoplasmic reticulum (ER) is a highly dynamic, variable, and extensive nuclear membrane-bound network in eukaryotic cells. It consists of two parallel membranes that form a tangled system of tubules and cisternae, and plays key roles in lipid metabolism and the biosynthesis and sorting of proteins within the cell. Additionally, in plant cells the ER mediates communication between the endomembrane system and non-secretory organelles, such as mitochondria, peroxisomes, and chloroplasts. Proteins in the ER undergo post-translational modifications, such as *N*- and *O*-glycosylation and hydrogen bond formation [1,2,3,4]. Studies conducted during the last two decades have also shown that the ER is involved in regulating the distribution of phytohormones and signalling via auxins [5,6,7,8,9], cytokinins [10,11], and ethylene [12].

We have focused on auxin (indole-3-acetic acid, IAA), a key phytohormone regulating a variety of crucial growth and developmental processes, and related metabolites. Recent findings indicate that the ER plays a central role in maintaining subcellular auxin homeostasis by regulating its biosynthesis, subcellular distribution, and metabolism, and also probably regulates its signalling [9]. In this way, the ER significantly affects the tightly balanced levels of auxin in plant cells and thus the growth and development of plant tissues and organs.

The subcellular redistribution of IAA is facilitated by a complex system of transporters including PIN-FORMED 5, 8 (PIN5, 8) and PIN-LIKES transporters (PILSs) located in the ER [5,6,7,8,13,14] (Figure 1), and the vacuolar transporter WALLS ARE THIN 1 (WAT1) [15]. The mechanisms regulating intracellular auxin distribution remain unclear, but the activity of ER-resident transporters directly affects auxin signalling [14,16]. Moreover, it is proven that the main IAA flux to nuclei goes through the ER [9].

Several IAA biosynthetic pathways in Arabidopsis have been reported [17,18]. However, the main pool of active IAA is predominantly synthesised via the TRYPTOPHAN AMINOTRANSFERASE OF ARABIDOPSIS/YUCCA (TAA/YUC) biosynthetic pathway [19,20]. The TAA/YUC complex resides in the ER membrane, but its catalytic domain faces the cytosol [21,22]. Other YUCCA members also co-localise with ER markers or are localised to the cytosol [18]. Newly synthesised IAA must be delivered to its site of action. Auxin perception systems that trigger transcriptional responses are found in the nucleus [23]. However, a key component of the nuclear IAA receptor complex, TRANSPORT INHIBITOR RESPONSE 1 (TIR1), is also detected in the cytosol, where it may mediate a rapid non-transcriptional response [24].

The distribution and levels of active IAA are strictly controlled and fine-tuned through complex coordination of its biosynthesis and transport, and by its inactivation [25,26]. Reversible inactivation is mediated via conjugation with sugars to form IAA-glucose (IAA-glc), which is catalysed by UDP-glucosyl transferases [27]. Another mode of inactivation is GRETCHEN HAGEN 3 (GH3)-catalysed conjugation of IAA with amino acids to form IAA-amino acid (IAA-aa) conjugates [28,29]. It is currently thought that the two most common IAA-aa forms, IAA-aspartate (IAAsp) and IAA-glutamate (IAGlu), cannot be converted back to free IAA [30,31]. However, less abundant IAA-aa forms, such as conjugates of alanine, leucine or valine, can be hydrolysed back to IAA by ER-localised amidohydrolases, leading to an increased local IAA concentration and thus increased signalling [32]. It has been suggested that IAA-aa are formed in the cytosol by GH3 [33] but hydrolysed in the ER [32]. In Arabidopsis, the dominant catabolic pathway responsible for reducing active IAA levels is regulated by the cytosolic enzyme DIOXYGENASE FOR AUXIN OXIDATION 1 (DAO1), which converts IAA into 2-oxindole-3-acetic acid (oxIAA) [26,34]. Additionally, oxIAA can be conjugated with glucose to form oxIAA-glucose (oxIAA-glc), the most abundant auxin metabolite in Arabidopsis [30,35].

Even though the localisations of many phytohormone-related enzymes, transporters and receptors are known, there is currently no comprehensive subcellular auxin map. Organelle-level auxin and cytokinin profiles have only been determined in vacuoles to date [15,36]. However, IAA was detected in chloroplasts during a cytokinin profiling study [37,38]. Subcellular phytohormone profiling is challenging due to the low abundance of plant hormones, the difficulty of isolating pure organelles, and the frequent use of organelle isolation buffers with high salt concentrations that interfere with subsequent MS-based analysis. Conventional organelle isolation methods generally rely on density gradient (ultra)centrifugation, which requires that organelles be gently and effectively released from plant tissue. This can be achieved by chopping with razor blades [39] or by enzymatic cell wall digestion [40,41,42]. Organelles are then separated based on the different velocities at which they move through a (dis)continuous sucrose density gradient [43], Ficoll [44], and/or Percoll [45,46]. Organelles become concentrated in the region where their density is equal to that of their surroundings (the so-called isopycnic point), where they stop moving through the gradient [47]. The migration velocity of organelles depends on their density, which in turn depends on their lipid/protein ratio, size and shape [48]. Alternative fractionation methods based on flow cytometry [39,49] or affinity purification have been developed more recently [50,51]. Interestingly, these methods enable the isolation of cell-type-specific organelles, such as nuclei [52] and mitochondria [53]. Despite the existence of advanced subcellular fractionation methods for well-bounded organelles, density-gradient centrifugation remains the gold standard for isolation of the ER, Golgi apparatus (GA), and vacuoles.

Here we present auxin metabolite profile of ER samples from Arabidopsis seedlings obtained using an optimised protocol based on density gradient ultracentrifugation (Figure 2 and Appendix A). Proteomic analysis revealed high ER enrichment in the isolated fractions, and showed that the optimised isolation protocol exhibits good reproducibility. The findings presented herein provide new insights into the subcellular distribution of auxin and IAA homeostasis.

## 2. Results

### 2.1. Isolation of the ER from Arabidopsis Plants

To analyse the ER auxin metabolite profile, it was first necessary to isolate a highly enriched ER fraction containing sufficient material for analytical purposes. Ding et al. [6] studied the mechanism of auxin transport in the ER and developed a method for isolating ER samples from Arabidopsis plants by centrifugation in a discontinuous sucrose density gradient. We adapted and optimised this protocol to increase the yield and purity of the product to a level sufficient for analytical determination of auxin metabolites within the ER.

The initial and the most critical step of the isolation process was the homogenisation of the plant material. The homogenisation method must provide a sufficient yield of the isolated compartment while avoiding its disintegration. We therefore compared homogenisation using a razor blade [6] to the method of homogenisation using a mortar and pestle with quartz sand, which was previously shown to be an effective homogenisation method for isolating intact mitochondria from Arabidopsis [45]. Gradient fractionation was then performed to assess the efficiency of organelle separation. During the optimisation process, the yields and purities of homogenates and the subsequently isolated ER fractions were evaluated by Western blot analysis, using organelle markers for the ER (Lumena-binding protein, BiP and Calnexin homolog 1/2, CNX1/2), nuclei (Histone 3, H3), Golgi complex (Coatomer subunit gamma, Sec21p) vacuoles (Epsilon subunit of tonoplast H+ATPase, V-ATPase), chloroplasts (D1 protein of photosystem II, PsbA), plastids (Glutamine oxoglutarate aminotransferase, GOGAT), mitochondria (H protein of glycine decarboxylase complex, GDC-H), cytosol (actin, ACT) and plasma membrane (plasma membrane H+ATPase, H-ATPase).

In general, the strength of all organelle marker signals was higher in the mortar–pestle homogenates than in the razor blade homogenates. However, for both homogenisation methods, the fraction expected to contain the ER (i.e., the fraction located at the 1.1/1.3 M sucrose layer interface) was only slightly enriched in ER markers (Figure 3a). Additionally, chloroplast, vacuole, and Golgi apparatus (GA) markers were detected in these fractions. However, only negligible mitochondrial and nuclear marker signals were detected by Western blotting in the fraction obtained after razor blade homogenisation. The presence of multiple marker signals in the putatively ER-enriched fractions indicates that organelles disintegrated during homogenisation with a mortar and pestle. Overall, these results show that razor blade homogenisation was the superior method for ER isolation (Figure 3a) and that chloroplasts or thylakoids were concentrated in the ER-enriched fraction along with the ER (Figure 3a).

To address the problem of unwanted organelles co-migrating with the ER, the preparation of samples for density gradient centrifugation was further optimised. First, the initial centrifugation step was optimised by performing centrifugation with centrifugal forces of between 2000 and 12,000× *g*. Western blot analyses of the resulting supernatants and the parent homogenate showed that the chloroplast content of the supernatant declined as the centrifugal force increased (Appendix A). However, chloroplasts were not fully removed from the samples during the initial centrifugation under any conditions. Based on the intensity of the chloroplast marker in the Western blot analysis, 4000× *g* was selected as the optimal centrifugal force. Importantly, the signal of the ER marker was not changed (Appendix A). Initial low-speed centrifugation eliminated nuclei from the final ER extract and greatly reduced its content of chloroplasts without appreciably affecting its ER content.

In the next step, the density gradient was optimised to eliminate residual chloroplasts and other contaminating organelles from the ER-enriched fraction (Figure 3b). Since chloroplasts should have a higher density than ER, the effect of reducing the density of individual layers of the sucrose gradient was tested. It was ultimately found that using a gradient with sucrose solution concentrations of 31%, 27%, 19%, and 8% (*w*/*w*) caused chloroplast sedimentation to the bottom of the tube, eliminating the chloroplast marker signal from the ER-enriched fraction without affecting the ER yield (Figure 3b). This adjustment of the gradient also affected the migration of vacuoles and GA, which had been additional contaminants of the ER-enriched fraction obtained with the original gradient (Figure 3a,c). Western blot analysis of the ER-enriched fraction isolated using the modified gradient revealed only a weak signal of the GA marker Sec21p (Figure 3c). Aside from the tonoplast marker V-ATPase, no other organelle markers were detected, and the degree of vacuole enrichment was much lower than when using the original gradient (Figure 3a). Optimising the isolation protocol by increasing the initial centrifugation speed and reducing the density of the gradient solutions thus greatly increased the purity of the ER-enriched fraction.

### 2.2. Confirmation of ER-Enriched Fractions by Proteomic Analysis

To validate our optimised ER isolation procedure, the retentate obtained by partitioning ER-enriched samples using Amicon^®^ filters was subjected to proteomic analysis. On average, 1015 proteins were unambiguously identified per sample, giving over 1300 individual identifications for five biological replicates (Figure 4a, Appendix A). Of these, 1003 proteins (about 75 % of the total identifications) were present in at least three replicates (Figure 4a), suggesting that the optimised isolation method exhibits high reproducibility. This was confirmed by pairwise comparisons of protein identifications, which indicated a mean reproducibility of 72.8 % (Appendix A) with a coefficient of variation of 4.6 %.

The ER enrichment of the prepared samples was investigated using two different approaches. First, we investigated the acquired dataset for the presence of proteins commonly used as organelle markers in Western blot experiments. The relative abundance of these markers was subsequently quantified based on the well-established intensity-based absolute quantification (iBAQ) intensities [54]. For comparative purposes, the same quantitative analysis was applied to control samples consisting of a total protein lysate prepared from 10-day-old Arabidopsis seedlings, which was analysed in the same way as ER-enriched isolates. As shown in Figure 4b, the detected ER markers were specifically enriched in the prepared isolates, while most markers for other organelles were significantly more abundant in the control samples. The only exceptions were the plasma membrane H+ATPase (AHA1) and vacuolar Epsilon subunit of tonoplast H+ATPase (VHA-E1; Figure 4b), consistent with the results obtained by Western blotting (Figure 3c). It should also be noted that some markers (e.g., the nuclei-specific H3) were detected only in the control samples.

Second, we performed a bioinformatic analysis of the ER-related dataset. Seven different algorithms were used to predict the ER localisation of the identified proteins, and the association of these proteins with ER-related processes was assessed based on functional annotation clustering of enriched gene ontology (GO) terms and KEGG pathways. The in silico predictions revealed 188 proteins potentially localised to the ER (Appendix A), representing over 14 % of the total identifications. Four of the five most enriched GO term clusters were directly connected to the ER, as shown in Figure 4c (Appendix A), which was consistent with the KEGG pathway analysis (Appendix A). Taken together, these results prove the ER enrichment of the isolated fraction, the high reproducibility of the modified ER isolation protocol based on a discontinuous sucrose density gradient, and its good compatibility with standard proteomic techniques.

### 2.3. Auxin Metabolite Determination in ER

Finally, the ER-specific auxin metabolome was determined in a filtrate containing low-molecular weight metabolites obtained by partitioning ER-enriched samples with Amicon^®^ filters. These filtrates were pure samples containing only minimal quantities of plant matrix. However, because the ER samples were isolated by density gradient centrifugation, they had a high content of sucrose, which could have adversely affected their analysis by liquid chromatography–tandem mass spectrometry (LC-MS/MS). Solid-phase extraction (SPE) was therefore used to eliminate sucrose from the samples and enrich the target analytes. To maximise extraction efficiency and auxin metabolite recovery, we tested two purification protocols previously developed to isolate IAA metabolites from plant tissue [55,56]. ER buffer samples containing sucrose concentrations ranging from 0.6 M to 1.2 M were spiked with a mixture of auxin standards (1 pmol each) and processed using either reversed-phase Oasis™ hydrophilic lipophilic balance (HLB) columns [55] or an in-tip micro solid-phase extraction (μSPE) method [56]. Higher extraction efficiencies were obtained using the μSPE method, for which the average recovery of all tested metabolites was around 60% over the range of tested sucrose concentrations, compared to 30% for the HLB-based method (Appendix A). The μSPE approach was therefore used to isolate IAA metabolites from the ER-enriched fractions prepared by sucrose density gradient centrifugation.

We next investigated the possibility that undesired changes in endogenous auxin levels might occur during the process of ER enrichment by ultracentrifugation. To this end, a total organelle suspension prepared from 10-day-old Arabidopsis seedlings was centrifuged at 4000× *g* and the resulting supernatant was incubated under identical conditions to those used during the ultracentrifugation step. Samples were collected at the beginning of incubation (0 h) and after 3 h, which corresponds to the duration of ER isolation by ultracentrifugation. Upon comparing the relative abundances of free IAA and auxin metabolites at the beginning and end of the incubation, we observed that the proportion of free IAA increased slightly from 2.3% to 2.8%, while that of IAA-glc decreased slightly from 2.1% to 1.8% (Figure 5a). Additionally, the relative abundance of oxIAA increased from 13.4% to 23% during the incubation, while that of oxIAA-glc decreased by 10%. The relative abundances of the IAA amide conjugates IAAsp and IAGlu remained unchanged during the incubation period (Figure 5a).

The optimised ER isolation procedure and the in-tip μSPE method were used together with LC-MS/MS to determine the metabolic profile of IAA in the ER. We first determined and compared the proportion of IAA metabolites in the crude Arabidopsis seedling extracts, the total organelle suspension, and the ER-enriched fraction (Figure 5b, Appendix A). The relative abundance of IAA in the total pool of analytes in the ER-enriched samples (8.8%) was five times that in the crude extract (1.7%). Conversely, the relative abundance of most IAA metabolites (IAAsp, IAGlu, IAA-glc and oxIAA) in the ER fraction was lower than in the crude extract and total organelle suspension. The main auxin metabolite in all three sample types was oxIAA-glc, whose relative abundance was relatively stable and ranged from 70 to 80% (Figure 5b). Finally, we used the total protein content of each sample (determined by MS) to normalise the levels of the analytes in order to compare the concentrations of IAA and its metabolites in the ER-enriched fraction with those in the crude Arabidopsis extract. The levels of individual IAA metabolites relative to the total protein content for each sample type are presented in Appendix A. Surprisingly, the ER-enriched fraction contained considerably higher levels of all studied analytes; levels of IAA metabolites were between 5 (IAAsp) and 12 (oxIAA-glc) times higher in the ER-enriched fraction than in the crude extract, and the level of free IAA in the ER was 62 times that in the crude extract (Figure 5c).

## 3. Discussion

To perform ER-specific proteomic and IAA-metabolomic analyses, it is essential to start with a highly pure ER fraction with minimal contamination by other organelles. Additionally, the ER fraction must be isolated using a method that does not destroy the organelle while simultaneously achieving sufficient enrichment for subsequent metabolite determination. Most published protocols for ER isolation were developed and optimised to isolate enriched ER membrane samples in order to study membrane proteins. The purity of the fractions obtained using these protocols is rarely reported, however.

With the aim of preparing an ER fraction suitable for studying the IAA metabolome in the ER, we adapted the ER isolation protocol of Ding et al. [6], which is based on ultracentrifugation in a discontinuous sucrose gradient. The individual steps of the protocol were optimised to maximise ER enrichment while minimising contamination by other organelles. Homogenisation with a mortar and pestle delivered a greater organelle yield than chopping the plants with a razor blade (Figure 3a) but was found to be unsuitable for our purposes because of the low purity of the final ER fraction. This low purity may be due to disintegration of organelle membranes caused by excessive homogenisation of plant tissue with the mortar and pestle, which prevents their fractionation [57]. It is also reasonable to assume that excessively rough homogenisation of plant material before ER isolation would cause leakage of the organelles’ contents. Chopping the material with a razor blade proved to be a gentler but still adequate method of homogenisation prior to ER isolation (Figure 3a).

Chloroplasts were the main contaminants of the ER fraction obtained using the original method. Therefore, the initial centrifugation of the total organelle suspension and the subsequent density gradient ultracentrifugation were optimised to remove chloroplasts from the final fraction (Figure 3b). The optimisation strategy exploited the fact that chloroplasts are denser than the ER [58]; consequently, reducing the density of the gradient solutions caused sedimentation of chloroplasts at the bottom of the centrifugal tube and yielded a purer ER fraction. The final fraction obtained using the modified protocol contained no visible chloroplast marker signal (Figure 3a). However, Western blot analysis of the ER-enriched fraction isolated using the optimised ultracentrifugation protocol revealed the presence of GA and vacuolar marker proteins. The GA and the ER have similar biochemical characteristics and are difficult to separate on the basis of density [59]. Despite this, the signal of GA marker protein Sec21p was greatly weakened in the ER fraction, which was consistent with the results of a proteomic analysis (Figure 4). Conversely, both Western blotting and MS-based protein analysis revealed the presence of the vacuolar marker protein V-ATPase in the ER-enriched fraction (Figure 3a and Figure 4c). However, this did not necessarily imply the presence of intact vacuoles in that fraction; although V-ATPase is a tonoplast-resident protein, it is synthesised in the ER and transported to the vacuole by intracellular trafficking [60]. It is thus possible that at least some of the V-ATPase signal was due to protein awaiting delivery to the vacuole rather than the presence of intact vacuoles in the ER fraction.

The successful analysis of the protein complement of our samples proved their ER enrichment and allowed us to draw several important conclusions. First, the high overlap of protein identifications in the analysed replicates indicated that our ER isolation protocol exhibits good reproducibility. This claim is greatly strengthened by the fact that the analysed samples were obtained from independently cultivated plants and independent ER isolations; in other words, they were true biological replicates. Second, the method allowed for plant ER enrichment. This was indicated not only by significant enrichment of established ER markers, such as BiP, CNX1, and calreticulin (CRT), but also by the ER-localisation of a substantial proportion (14 %) of the identified proteins. This proportion of ER-localised proteins was very high, given that ER proteins typically comprise about 1% of the total dataset in standard proteomic experiments [61]. As might be expected, this was accompanied by an overrepresentation of cellular processes associated with the ER including protein processing and post-translational modification as well as biosynthesis of secondary metabolites and phenylpropanoids [62]. Finally, extracts prepared using the optimised protocol were highly compatible with proteomic methods, and could thus provide important new and detailed information on the ER and its homeostasis. It also seems plausible that these extracts would be compatible with other modern “omics” techniques.

IAA metabolite profile analyses revealed considerable differences in the relative abundance of active IAA and its most abundant metabolites when comparing the ER-enriched fraction to other plant extracts. Specifically, free IAA accounted for a greater proportion of the total IAA metabolite pool in the ER fraction than in crude plant extracts or total organelle suspensions, and the opposite was true for most other IAA metabolites (Figure 5b). Control experiments were performed to determine the extent to which these differences in metabolite distributions could be attributed to undesirable metabolic transformations occurring during the relatively lengthy isolation of the ER fraction. It was found that small changes in the relative abundance of some metabolites did occur during the incubation of organelle suspension—specifically, the relative abundance of free IAA and oxIAA increased, while that of IAA-glc and oxIAA-glc decreased, suggesting that some hydrolysis of IAA glucosyl esters occurred during sample preparation (Figure 5a). However, the relative abundance of IAA only increased by 0.5 percentage points during the control experiment; ER enrichment caused a much greater increase of four percentage points when compared to the total organelle suspension, and seven percentage points relative to the crude extract (Figure 5b). Additionally, whereas the relative abundance of oxIAA increased by 10 percentage points in the control experiments, its relative abundance in the ER-enriched fraction fell by five percentage points (Figure 5a,b). This suggests that the observed differences in the distribution of auxin metabolites were mainly due to the ER enrichment of the obtained fraction rather than undesirable metabolic changes during the ER isolation procedure.

The relative abundances of the various IAA metabolites in each sample type were also normalised against the samples’ total protein contents, revealing that the absolute concentrations of IAA and all its metabolites in the ER-enriched fraction were substantially higher than in the crude whole-plant extract. Interestingly, this difference was most pronounced for free IAA (Figure 5c). One might reasonably expect IAA in its active form to accumulate in the ER and to eventually be transported to the nucleus, where the auxin signal is perceived and the signaling pathway is triggered, as suggested by Middleton et al. [9]. The lower relative abundance of most IAA metabolites in the ER (when compared to whole-plant and organelle extracts) may suggest that the formation and accumulation of IAA metabolites and conjugates occurs predominantly in other cell compartments. For example, analysis of the vacuolar IAA metabolite profile revealed that most of the total IAA in that organelle exists as the glycosyl ester (IAA-glc), a storage form that can be re-hydrolysed to the active form [15].

## 4. Materials and Methods

### 4.1. Plant Material and Growth Conditions

Ten-day-old *Arabidopsis thaliana* ecotype Col-0 were used as the material for experiments. Seeds were surface-sterilised using a 70% ethanol solution (Merck Life Science, Darmstadt, Germany) supplemented with 0.1% Tween-20 (Merck Life Science, Darmstadt, Germany) for 10 min, rinsed with sterile deionised water, and sowed on Murashige and Skoog solid media supplemented with 1% sucrose. After 3 days of stratification, the plates were arranged vertically and incubated under long-day conditions (16 h light/8 h dark) at 22 ± 1 °C.

### 4.2. Homogenisation

For the homogenisation process, 2 g (FW) of whole Arabidopsis seedlings were homogenised in 2 mL of ice-cold homogenisation buffer (0.5 M sucrose, 5 mM MgCl_2_∙6H_2_O, 0.1 M KH_2_PO_4_, 1 mM 1,4-Dithiothreitol, 1 mM phenylmethylsulfonyl fluoride, 1 mM Roche cOmplete™ Protease Inhibitor Cocktail EDTA-free, pH 6.65; all components from Merck Life Science, Darmstadt, Germany). The plant material was transferred to a petri dish placed on ice. All subsequent steps were performed on ice using precooled solutions and implements. The sample was chopped using a razor blade for 5 min, or homogenised in a mortar containing quartz sand with a pestle and incubated for 5 min. The resulting homogenate was filtered into a 50 mL falcon tube through two layers of Miracloth pre-wetted with homogenisation buffer. Residual homogenised material was washed out of the petri dish or mortar with 4 mL of homogenisation buffer. An aliquot of the resulting total organelle suspension (500 μL) was collected and immediately frozen in liquid nitrogen and stored at −80 °C until Western blot or LC-MS/MS analysis. ER enrichment was then performed according to the original protocol [6] or the optimised protocol as described below.

### 4.3. Optimisation of Initial Centrifugation

Arabidopsis seedlings were homogenised with a razor blade as described above. The filtrate was then centrifuged at 2000× *g*, 3000× *g*, 4000× *g*, 5000× *g*, 6000× *g*, 7000× *g*, 9000× *g*, or 12,000× *g* (10 min, 4 °C), after which 1 mL of supernatant (S) was collected and immediately frozen in liquid nitrogen and stored at −80 °C until Western blot analysis.

### 4.4. Preparation of ER-Enriched Fraction–Original Protocol

The density gradient centrifugation protocol was adapted from that of Ding at al. [6]. The sucrose solutions used to establish the gradients were prepared by dissolving sucrose in the ER buffer (5 mM MgCl_2_∙6H_2_O; 0.1 M KH_2_PO_4_; pH 6.65). The supernatant obtained after initial centrifugation at 4000× *g* (S4000) was then divided into two aliquots of approx. 3 mL each. Each aliquot was then carefully loaded on top of a 3 mL ice-cold 1.3 M sucrose cushion in an ultracentrifuge tube to establish a two-step density gradient, after which the aliquot was supplemented with a further 3 mL of homogenisation buffer to prevent tube collapse. The two tubes were then loaded symmetrically into the centrifuge and spun at 108,000× *g*, 4 °C for 90 min. After ultracentrifugation, the upper phase was removed without disrupting the focused microsomal fraction, which was then slowly overlaid with 3 mL of 1.1 M, 3 mL of 0.7 M, and 3 mL of 0.25 M sucrose solutions. The resulting four-step gradients were centrifuged at 108,000× *g*, 4 °C for 90 min. The ER-enriched fraction located at the 1.1/1.3M interphase (1 mL) was then transferred into a new tube and frozen for phytohormone and protein extraction. Alternatively, collected fractions were ultracentrifuged again at 108,000× *g* and 4 °C for 50 min to concentrate the ER-enriched fraction. The supernatant was then removed and the pellet was resuspended in 50 µL of homogenisation buffer. The resulting ER-enriched fraction was immediately frozen in liquid nitrogen and stored at −80 °C until Western blot analysis was performed.

### 4.5. Preparation of ER-Enriched Fraction-Optimised Protocol

The two- and four-step gradient centrifugation processes were performed as described above, but the concentrations of the gradient-forming sucrose solutions were slightly reduced to 31%, 27%, 19%, and 8% (*w*/*w*). To determine the IAA metabolite profile of the ER, 1 mL of the ER-enriched fraction located at the interface of the 27% and 31% sucrose solutions was taken and processed as described above.

### 4.6. The SDS-PAGE Western Blot Assay

The samples were mixed with Laemmli Sample Buffer (Bio-Rad, Hercules, CA, USA) supplemented with 10% β-mercaptoethanol (Merck Life Science, Darmstadt, Germany) and incubated in a thermoblock at 70 °C for 5 min. Samples were then centrifuged in a MiniSpin^®^ centrifuge (Eppendorf, Hamburg, Germany) at 10,000 rpm for 5 min before separation of the protein mixtures on a 12% polyacrylamide gel (Bio-Rad, Hercules, CA, USA). Dual Color Standards (Bio-Rad, Hercules, CA, USA) was used as a molecular weight marker. Electrophoresis was performed first at 90 V for 30 min and then at 120 V until the end of the separation. After protein migration, the gel was rinsed for 5 min in transfer buffer (150 mM), and proteins were transferred for 2 h at 290 mA and 4 °C onto 0.45 µm nitrocellulose membranes (Santa Cruz Biotechnology, Heidelberg, Germany). The membranes were then blocked with 5% low-fat milk in TBS-T for 1 h, cut into segments, and incubated for 1 h with rabbit primary antibodies (Agrisera, Vännäs, Sweden) against the organelle markers listed below: anti-BiP (1:2500; AS09 481), anti-CNX1/2 (1:2500; AS12 2365), anti-V-ATPase (1:2000; AS07 213), anti-H3 (1:5000; AS10 710), anti-Sec21p (1:1000; AS08 327), anti-PsbA (1:10,000; AS05 084), anti-GDC-H (1:5000; AS05 074), anti-GOGAT (1:1000; AS07 242), anti-ACT (1:2500; AS13 2640) and anti-H-ATPase (1:1000; AS07 260). All primary antibodies were diluted in 1% low-fat milk in TBS-T. Membrane segments were then incubated with goat anti-rabbit IgG (H&L) HRP-conjugated secondary antibody (1:10,000; AS09 602) diluted in 1% low-fat milk in TBS-T for 1 h. Visualisation was performed with a chemiluminescence kit (SuperSignal™ West Pico Chemiluminescent Substrate (Thermo Fisher Scientific, Waltham, MA, USA) using a ChemiDoc MP Imaging System (Bio-Rad, Hercules, CA, USA).

### 4.7. Proteomic Analysis

Proteins in the retentate from sample partitioning on Amicon^®^ Ultra centrifugal filter units (cut-off: 3 kDa) (Merck Millipore, Burlington, MA, USA) were precipitated with ice-cold acetone, recovered by centrifugation and digested *in solution* with commercially available trypsin as previously described [63]. Five biological replicates were processed in this way. As control samples, total protein extracts were prepared from 10-day-old Arabidopsis seedlings as described by Basal et al. [64] and digested in solution in the same way as the ER isolates (see above). Six control biological replicates were prepared and analysed. The tryptic peptides were purified on a home-made reversed-phase (C18) microcolumn according to Franc et al. [65] and analysed by LC-MS/MS using settings adapted from Chamrád et al. [66]. The collected MS data were processed and searched using MaxQuant software, version 1.6.17.0 [67] with the “Bruker QTOF” instrument parameter setting [68] and the Andromeda engine [69]. Protein identification was achieved using the *Arabidopsis thaliana* (cv. Columbia) protein database (UniProt, reference proteome UP000006548, 39,345 protein sequences, downloaded 8 March 2021) supplemented with 247 common laboratory contaminants. The iBAQ [54] was calculated to assess the relative abundances of the selected marker proteins, and the localisation of these proteins was investigated in silico using following predictors: BaCelLo [70]; iPSORT [71]; PProwler 1.2 [72]; PredSL [73]; SLPFA [74]; SLP-Local [75]; and TargetP 1.1 [76]. A consensus of at least four predictors was required for a protein to be assigned as ER-located. The enriched clusters of functional annotation GO terms and KEGG pathways related to the identified proteins were determined using DAVID Bioinformatics Resources 6.8 [77].

The total protein content of the samples was calculated from the MS data by integrating the area under the curve of the corresponding chromatogram. To this end, a series of protein digests with preset protein contents prepared from Arabidopsis seedlings were used for calibration.

All proteomics data were deposited with the ProteomeXchange consortium (http://proteomecentral.proteomexchange.org) via the PRIDE partner repository [78] with the dataset identifier PXD027522.

### 4.8. Optimisation of SPE Protocols

Sucrose solutions of 1.2 M, 0.8 M and 0.6 M prepared by dissolving sucrose in the ER-isolation buffer were divided into equal portions, which were supplemented with a mixture of auxin standards comprising IAA, IAA-glc, IAAsp, IAGlu, oxIAA and oxIAA-glc (1 pmol each). Some aliquots of each spiked sucrose solution were purified by SPE using Oasis™ HLB columns (30 mg/mL, Waters) following the protocol of Novák et al. [55]. Other aliquots were processed by in-tip μSPE [56]. Eluates were evaporated to dryness in vacuo and stored at −20 °C until LC-MS/MS analysis.

### 4.9. Control of Auxin Metabolite Profile Stability

As described above, 2 g (FW) of whole Arabidopsis seedlings were homogenised, filtered, and centrifuged at 4000× *g*. The supernatant was divided into 1 mL aliquots and transferred into microtubes. Half of the samples were immediately frozen in liquid nitrogen and stored at −80 °C. The other half were placed in a refrigerator (4 °C) and incubated for 3 h. At the end of the incubation, these samples were frozen in liquid nitrogen and stored at −80 °C until SPE extraction.

Samples were slowly thawed on ice and centrifuged (15 min, 4 °C, 21,000× *g*), after which the following stable isotope-labelled internal standards were added to each sample: [^13^C_6_]IAA, [^13^C_6_]oxIAA, [^13^C_6_]oxIAA-glc, [^13^C_6_]IAA-glc, [^13^C_6_]IAAsp a [^13^C_6_]IAGlu (5 pmol per sample). The samples were then purified by the SPE protocol using Oasis™ HLB columns [55]. Finally, the eluates were evaporated to dryness in vacuo and stored at −20 °C until LC-MS/MS analysis.

### 4.10. Extraction and Purification of IAA Metabolites

Samples of enriched ER fractions were slowly thawed on ice. The following stable isotope-labelled internal standards were added to each sample: [^13^C_6_]IAA, [^13^C_6_]oxIAA, [^13^C_6_]oxIAA-glc, [^13^C_6_]IAA-glc, [^13^C_6_]IAAsp and [^13^C_6_]IAGlu (5 pmol per sample). Amicon^®^ Ultracentrifugal filters (cut-off: 3 kDa) were used to separate proteins from low molecular weight substances including auxin metabolites. Filtrates were purified by in-tip µSPE as described by Pěnčík et al. [56]. Each filtrate (~3 mL) was divided into two equal parts, which were processed as independent replicates. Samples were acidified to pH 2.7 with 1 M hydrochloric acid, and two 500 μL aliquots were loaded onto a multi-StageTip column that had been activated with 50 μL of acetone (by centrifugation at 2200 rpm, 10 min, 4 °C), 50 μL of methanol (2200 rpm, 10 min, 4 °C), and 50 μL of redistilled water (2200 rpm, 15 min, 4 °C). The column was then washed with 50 μL of 0.1% acetic acid (3400 rpm, 15 min, 4 °C) and eluted with 50 μL of 80% methanol (3400 rpm, 15 min, 4 °C). Eluates from three columns were combined and evaporated to dryness in vacuo and stored at −20 °C until LC-MS/MS analysis. For quantification of IAA and its metabolites in Arabidopsis seedlings, samples containing 10 mg plant material (fresh weight) were extracted in 1 mL ice-cold sodium phosphate buffer (50 mM, pH 7.0, 4 °C) containing 0.1% diethyldithiocarbamic acid sodium salt. The mixture of internal standards (5 pmol per sample) was added to each sample. The samples were homogenised, extracted at 4 °C with continuous shaking (10 min), centrifuged (15 min, 21,000× *g* at 4 °C) and purified by in-tip µSPE as described above. Eluates were evaporated to dryness in vacuo and stored at −20 °C until LC-MS/MS analysis.

### 4.11. Quantification of IAA Metabolites

The evaporated samples processed by in-tip µSPE were dissolved in 30 µL of 10% methanol. Samples processed by SPE on Oasis™ HLB columns (Waters Corp., Milford, CT, USA) were dissolved in 40 µL of 10% methanol. All samples were mixed, sonicated for 5 min, and filtered using a Micro-spin^®^ filter tube (0.2 μm pore size; 3 min at 8000 rpm, (Chromservis, Praha, Czech republic). Determination of auxin metabolites was performed using a high-performance liquid chromatography–electrospray tandem mass spectrometry with a 1260 Infinity II HPLC system (Agilent Technologies, Santa Clara, CA, USA) equipped with a reversed-phase column (Kinetex; 50 mm × 2.1 mm, 1.7 µm; Phenomenex) coupled to a 6495 Triple Quad detector (Agilent Technologies, Santa Clara, CA, USA). Individual analytes were detected in positive and negative ion mode using optimised conditions [56].

## 5. Conclusions

Although auxins were the first phytohormones to be discovered [79], they continue to be studied intensively. While the subcellular partitioning of auxin biosynthesis, signalling, storage, and deactivation processes suggests the existence of complex mechanisms for maintaining auxin homeostasis, the distribution of IAA and its metabolites within the plant cell remains largely unknown. By combining organelle separation by density gradient centrifugation with ultrasensitive mass spectrometry-based analysis, it may be possible to perform detailed organelle-level auxin profiling to shed light on this issue [80].

To this end, we developed an improved protocol for ER isolation from Arabidopsis seedlings to determine the content of auxin and its metabolites in this organelle. Herein we present the first reported auxin metabolite profile in a highly ER-enriched fraction. We found that active IAA was substantially more abundant in the ER than in total plant extracts, which is consistent with the hypothesised importance of ER in auxin metabolism and signalling modulation [9]. In addition, we were able to characterise the protein content of the isolated ER fraction, confirming its enrichment with the desired organelle. Our improved ER isolation method could potentially enable further study of this organelle using other “omics” techniques, as well as more detailed studies on intracellular auxin transport.

## Figures and Tables

**Figure 1 ijms-22-09370-f001:**
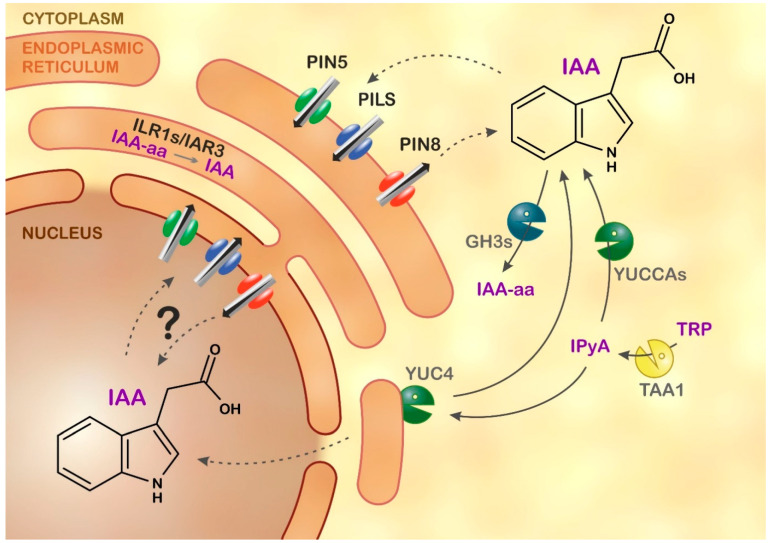
Model of auxin homeostasis in Arabidopsis endoplasmic reticulum (ER). Levels of active indole-3-acetic acid (IAA) are tightly regulated via biosynthesis, transport, and metabolism. Tryptophan (TRP) is converted to IAA via the TRYPTOPHAN AMINOTRANSFERASE OF ARABIDOPSIS/YUCCA (TAA/YUC) biosynthetic pathway, in which a key intermediate is indole-3-pyruvic acid (IPyA). Members of the PIN family of auxin transporter proteins (PIN5, PIN8) and PIN-LIKES (PILS) facilitate intracellular auxin transport. PIN5 mediates auxin flux from the cytosol to the ER lumen, whereas PIN8 acts in the opposite direction. IAA is inactivated by Gretchen Hagen 3 (GH3) proteins, which catalyse the formation of IAA-amino acid conjugates (IAA-aa). ER-localised auxin amidohydrolases (IAR3, ILL2 and ILR1) catalyse the reverse reaction, hydrolysing IAA-aa to active IAA. The ER membrane is in light brown, while the nuclear membrane is in dark brown. Solid and dotted arrows indicate enzymatic conversion or transport, respectively. “**?**” means putative auxin transport.

**Figure 2 ijms-22-09370-f002:**
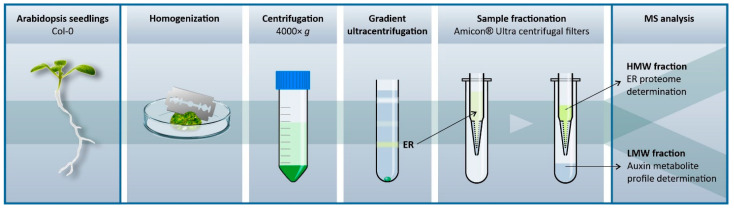
ER isolation workflow. 10-day-old Arabidopsis seedlings were placed in a Petri dish on ice and finely chopped with a razor blade in 2 mL of ice-cold homogenisation buffer. The resulting homogenate was filtered through two layers of Miracloth into a 50 mL tube and centrifuged at 4000× *g* and 4 °C for 10 min. Organelles in the resulting supernatant were fractionated using a discontinuous sucrose density gradient. The ER-enriched fraction was then passed through Amicon^®^ Ultra centrifugal filters (cut-off: 3kDa) to obtain a high-molecular weight (HMW) fraction containing proteins and a low-molecular weight (LMW) fraction for auxin metabolite determination by liquid chromatography coupled with tandem mass spectrometry.

**Figure 3 ijms-22-09370-f003:**
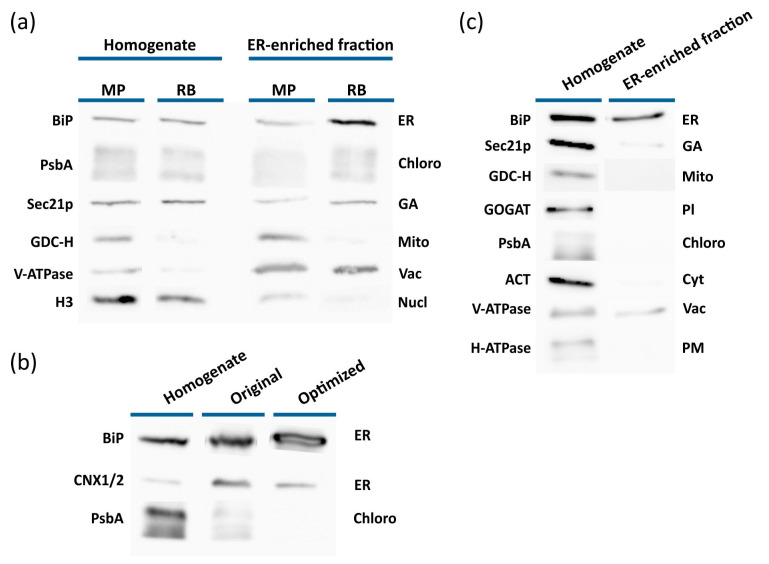
Optimisation of ER isolation from 10-day-old seedlings of *A. thaliana*. The effectiveness and usefulness of each step in the optimisation process was verified by Western blot analysis. (**a**) Comparison of two homogenisation methods–grinding seedlings using a mortar and pestle (MP) with added quartz sand, and chopping with a razor blade (RB). In both cases, the ER-enriched fraction was isolated using the originally reported sucrose density gradient [6]. (**b**) Organelle profiles of ER-enriched fractions obtained using various density-gradient ultracentrifugation protocols. Organelles were released by chopping seedlings with razor blade. Microsomal fractions were separated using the original or optimised (reduced sucrose density) gradients; for details, see Materials and Methods, Section 4.5 and Section 4.6. (**c**) Final evaluation of the ER-enriched fraction obtained using the optimised sucrose density gradient. The following organelle-specific markers were immunodetected: Endoplasmic reticulum–ER (Lumena-binding protein, BiP and Calnexin homolog 1/2, CNX1/2), nuclei–Nucl (Histone 3, H3), Golgi apparatus–GA (Coatomer subunit gamma, Sec21p) vacuoles–Vac (Epsilon subunit of tonoplast H+ATPase, V-ATPase), chloroplasts–Chloro (D1 protein of photosystem II, PsbA), plastids–Pl (Glutamine oxoglutarate aminotransferase, GOGAT), mitochondria–Mito (H protein of glycine decarboxylase complex, GDC-H), cytosol–Cyt (actin, ACT), and plasma membrane–PM (plasma membrane H+ATPase, H-ATPase).

**Figure 4 ijms-22-09370-f004:**
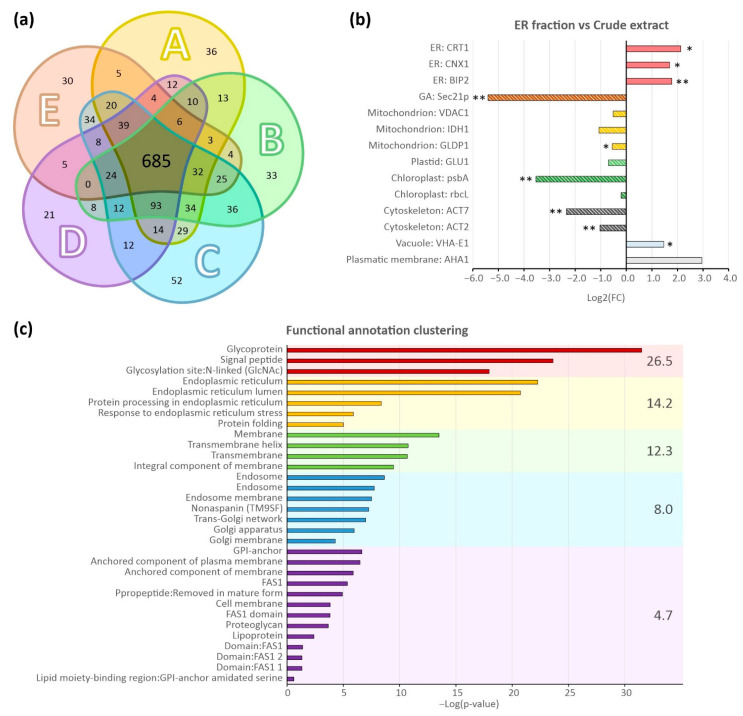
Proteomic analysis of the prepared ER fractions. (**a**) Protein identification overlaps for the analysed independent ER isolates (*n* = 5). The capital letters (A,B,C,D,E) stand for the individual ER isolation replicates. Numbers show the total sums of protein identifications belonging to the particular sections of the Venn diagram. (**b**) Distribution of Log2 fold changes (FC) of known protein organelle markers. Fold changes were calculated from the ratios of the iBAQ values for the respective markers identified in the analysed ER isolates (*n* = 5) to those in control samples (total protein lysates from 10-days old Arabidopsis seedlings; *n* = 5). Asterisks denote significant differences (* for *p* ≤ 0.05 and ** for *p* ≤ 0.01). (**c**) Clustering of significantly enriched functional annotation terms for the identified proteins whose location was assigned as the ER. The colours of bars and the background indicate the annotation cluster of the protein. The clusters are sorted according to their enrichment score that is presented on the right side of the graph, and the individual functional terms are sorted by the negative log of their *p*-value. Endoplasmic reticulum (ER), Golgi apparatus (GA).

**Figure 5 ijms-22-09370-f005:**
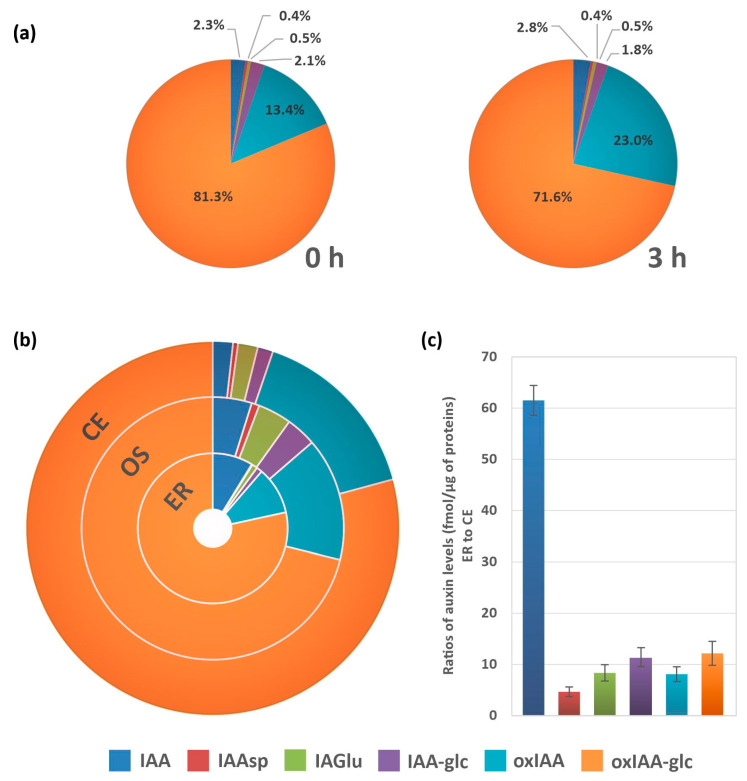
Determination of auxin metabolites in the Arabidopsis endoplasmic reticulum (ER). (**a**) Control of auxin metabolite profile stability. A homogenate prepared from 10-day-old seedlings was filtered and then centrifuged. The supernatant was immediately frozen or incubated in refrigerator at 4 °C for 3 h, as in the ER isolation process. Auxin metabolite profiles are expressed in percentages showing the relative abundance of each metabolite (*n* = 3). (**b**) Relative distribution of auxin metabolites in crude Arabidopsis extracts from 10-day-old seedlings, a total organelle suspension, and the ER-enriched fraction. (*n* = 5). (**c**) Abundance of auxin metabolites in ER. The enrichment of analytes is expressed as the ratio of the absolute level of each metabolite (fmol/µg of proteins) in the ER to that in the crude extract (*n* = 5). Error bars indicates. Crude extract (CE), organelle suspension (OS), Indole-3-acetic acid (IAA), IAA-aspartate (IAAsp), IAA-glutamate (IAGlu), IAA-glucose (IAA-glc), 2-oxoindole-3-acetic acid (oxIAA), oxIAA-glucose (oxIAA-glc).

## Data Availability

The data presented in the current study are available in the article and Appendix A.

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
