# Peer review of "Auxin Metabolome Profiling in the Arabidopsis Endoplasmic Reticulum Using an Optimised Organelle Isolation Protocol"

_ijms, 2021, doi:10.3390/ijms22179370_

Round 1

Reviewer 1 Report

The manuscript by Včelařová L, Skalický V, Chamrád I, Lenobel R, Kubeš MF, Pěnčík A and Novák O entitled “Revealing auxin metabolome in the endoplasmic reticulum of Arabidopsis thaliana” presents an optimized protocol for ER isolation from Arabidopsis seedlings and how this protocol could be used to determine ER-specific auxin metabolite profiles by high-performance liquid chromatography-electrospray tandem mass spectrometry.

The methods involved:

  1. The isolation of ER from Arabidopsis seedlings by comparing to homogenization techniques: Mortar/pestle and razor blade and results in figure 3a demonstrated that razor blade technique was superior shown by the strong signal in the ER- enriched fraction;
  2. Centrifugation was at 4000 g to remove nuclei and reducing the content of chloroplasts in ER content;
  3. Density gradient centrifugation by using different sucrose solution concentrations which optimized the ER signal in the western blot analysis in Figure 3b and key markers in Figure 3c.

Once the authors optimized the method for ER isolations, samples were subjected to proteomic and IAA metabolomic analyses with Amicon filters. Proteomic analysis identified genes in key functional groups (figure 4) and metabolomic analysis identified different types of IAA and free-IAA in ER was 62 times that in the crude extract (figure 5).

This is an interesting finding if one relates to Figure 1 where the authors presented a current model for auxin homeostasis in Arabidopsis ER. This could be mentioned in the conclusion.

Overall, the paper was well written.

Here are some recommendations:

On page 2 in figure 1, ER membrane have two different colors. Please explain in the legend. Is this due to being part of the nuclear membrane?

On page 5 line 167; please specify with part of the Materials and Methods for ease- ie. Materials and methods section 4.6

Reviewer 2 Report

In this study, the authors optimized ER isolation and established analysis of proteome and auxin metabolites in Arabidopsis. The reviewer thinks this is an interesting technique. To improve the manuscript, the reviewer proposes some points to be revised.

1) The authors are focusing on auxin metabolites in the title. Optimization of ER isolation is also important topic in the paper. Include that in the title and change to the appropriate title.

2) One question about auxin analysis. Can the authors analyze IAA precursors, Trp, IPA, IAOx in the same sample? Amount of IAA precursors is also important information.

3) L.182 How do the authors determine the initial centrifugation speed 4000 xg? Please explain that.

Minor points

1) L.80 IAA aas > IAA-ass

2) L.177, 182 and others, 12000 g, 4000 g > 12000 xg, 4000 xg

3) L.215 and L.227 Life stage of Arabidopsis seedlings are different. 10-days-old at L.215, 11-days-old at L227. Different conditions?

4) L.605 and 656 Year of reference papers. Not in bold.

5) Please let native researchers check English grammar in the text.
